# A spliced latency-associated VZV transcript maps antisense to the viral transactivator gene 61

Daniel P. Depledge [1,7], Werner J.D. Ouwendijk[2], Tomohiko Sadaoka [3], Shirley E. Braspenning[2], Yasuko Mori[3], Randall J. Cohrs[4,5], Georges M.G.M. Verjans[2,6] & Judith Breuer [1]

Varicella-zoster virus (VZV), an alphaherpesvirus, establishes lifelong latent infection in the neurons of >90% humans worldwide, reactivating in one-third to cause shingles, debilitating pain and stroke. How VZV maintains latency remains unclear. Here, using ultra-deep virus-enriched RNA sequencing of latently infected human trigeminal ganglia (TG), we demonstrate the consistent expression of a spliced VZV mRNA, antisense to VZV open reading frame 61 (ORF61). The spliced VZV latency-associated transcript (VLT) is expressed in human TG neurons and encodes a protein with late kinetics in productively infected cells in vitro and in shingles skin lesions. Whereas multiple alternatively spliced VLT isoforms (VLT$_{ly}$) are expressed during lytic infection, a single unique VLT isoform, which specifically suppresses ORF61 gene expression in co-transfected cells, predominates in latently VZV-infected human TG. The discovery of VLT links VZV with the other better characterized human and animal neurotropic alphaherpesviruses and provides insights into VZV latency.

[1] Division of Infection and Immunity, University College London, London WC1E 6BT, UK. [2] Department of Viroscience, Erasmus Medical Centre, 3015 CN Rotterdam, The Netherlands. [3] Division of Clinical Virology, Center for Infectious Diseases, Kobe University Graduate School of Medicine, 7-5-1 Kusunoki-cho, Chuo-ku, Kobe 650-0017, Japan. [4] Department of Neurology, University of Colorado School of Medicine, Aurora, CO 12700, USA. [5] Department of Immunology & Microbiology, University of Colorado School of Medicine, Aurora, CO 12800, USA. [6] Research Centre for Emerging Infections and Zoonoses, University of Veterinary Medicine Hannover, 30559 Hannover, Germany. [7] Present address: Department of Microbiology, New York University, New York, NY 10016, USA. These authors contributed equally: Daniel P. Depledge, Werner J. D. Ouwendijk, Tomohiko Sadaoka. These authors jointly supervised the work: Georges M. G. M. Verjans, Judith Breuer. Correspondence and requests for materials should be addressed to G.M.G.M.V. (email: g.verjans@erasmusmc.nl) or to J.B. (email: j.breuer@ucl.ac.uk)

During primary infection, neurotropic alphaherpesviruses (αHVs) gain access to neurons in sensory, cranial and autonomic ganglia to establish a lifelong latent infection from which they can reactivate to cause debilitating disease[1]. For the best-studied αHVs, including herpes simplex virus types 1 and 2 (HSV-1 and HSV-2), pseudorabies virus and bovine herpesvirus 1 (BHV-1), viral latency has been shown to be accompanied by the expression of a single or restricted set of latency-associated transcripts (LATs)[2–6]. These transcripts map antisense to the gene encoding the conserved multifunctional HSV infected cell protein 0 (ICP0), to which varicella-zoster virus (VZV) open reading frame 61 (ORF61) is homologous, which besides inhibiting intrinsic cellular antiviral responses is the major transcriptional transactivator of lytic viral genes required

for reactivation of latent HSV[2–10]. While the function of the LATs remains uncertain (reviewed in ref. [11]), there is mounting evidence from work on HSV-1, HSV-2 and BHV-1 that LATs induce generalized transcriptional and/or translational repression[12], and several studies have shown that LAT-encoded miRNAs (e.g., miR-H2, in HSV-1) or proteins (e.g., BHV-1 latency-related protein) target viral mRNAs including ICP0[13–15].

The exception has been VZV, a human-restricted αHV and causative agent of varicella and herpes zoster, for which no latency transcript mapping antisense to its ICP0 homologue ORF61 has been described[1,16–18]. While the absence of a canonical latency transcript in VZV may represent a fundamental difference in the evolution and biology of this virus, it is notable that a putative LAT, antisense to ORF61, has also been described for simian varicella virus (SVV), the varicellovirus most closely related to VZV. However, neither the transcript nor its function in SVV infection have been studied in detail[7,8].

Like other herpesviruses, lytic VZV infection is characterized by full viral gene expression occurring with temporally linked immediate-early (IE), early (E) and late (L) kinetics to generate infectious virus progeny[19,20]. By contrast, VZV gene expression during latency remains poorly defined[16,18,21–23]. This is largely due to the lack of appropriate animal[24] and, until recently, in vitro[25,26] models, which accurately mimic VZV pathogenesis[24–26]. VZV latency has been extensively studied in cadaveric human trigeminal ganglia (TG), a prominent anatomic site of both HSV-1 and VZV latency, yielding conflicting results as to which VZV transcripts and proteins are expressed[27]. Whereas viral protein detection by immunohistochemistry (IHC) can largely be attributed to non-specific binding of anti-VZV antibody preparations[23,28], the time interval between death and TG specimen processing (post-mortem interval, PMI) determines the number and quantity of VZV transcripts detected[16,18,22]. Using PCR primers targeting all canonical VZV genes, only the lytic gene ORF63 is occasionally detected in TGs with PMI < 9 h[18], whereas multiple viral transcripts of different kinetic classes are detected in human TGs with PMI > 9 h[16,18,22] and also frequently in animal models[29,30].

Here we describe the identification of a VZV latency-associated transcript (VLT), consistently detected in VZV and HSV-1 co-infected human TG that lies antisense to ORF61. Although multiple alternatively spliced transcripts are present during productive infection, the unique VLT isoform that supresses ORF61 gene transcription in co-transfected cells predominates during latency. The discovery of VLT unifies the VZV latent viral transcription programme with that of other better-characterized human and animal neurotropic αHVs while removing long-standing barriers to understanding VZV latency.

## Results
### Exclusive detection of HSV-1 LAT and miRNAs in human TG.
We first validated our experimental approach by sequencing enriched and unenriched viral transcripts from lytically VZV- or HSV-1-infected human retinal pigmented epithelial (ARPE-19) cells to demonstrate detection of all currently annotated VZV and HSV-1 genes at high specificity and sensitivity (Figs. 1 and 2 and Supplementary Fig. 1). RNA-Seq of unenriched RNA libraries from two latently VZV and HSV-1 co-infected human TGs (donors 1 and 2, Supplementary Tables 1 and 2), enriched for polyadenylated transcripts or ribosomal RNA-depleted total RNA (Supplementary Fig. 2), confirmed the presence of the stable 1.5/2.0-kb LAT-derived introns, the hallmark of HSV-1 latency[6]. Enrichment for polyadenylated HSV-1 sequences in the same TGs and five additional dually latently infected TGs (donors 1–7, Supplementary Tables 1 and 2) revealed both LAT introns and the near-complete 8.3-kb full-length LAT transcript from which they derive but no other viral transcripts (Fig. 1 and Supplementary Fig. 3). The latency-associated miRNAs (mir-H2, mir-H3, mir-H4, mir-H6, mir-H7 and mir-H14)[13] were also detected in three TGs (donors 1, 3 and 5) selected for and analysed by miRNA sequencing of unenriched RNA libraries (Supplementary Fig. 4). These data illustrate the high specificity and sensitivity of our target-enriched RNA-Seq methodology and clearly demonstrate that the HSV-1 latency transcriptome in human TG is limited to the LATs and encoded miRNAs.

### Identification of a spliced VZV latency-associated transcript.
While no VZV transcripts could be identified in non-enriched TG RNA samples (Supplementary Fig. 5), enrichment for VZV sequences in polyadenylated RNA revealed the presence of a novel 496-nucleotide multi-exon transcript located antisense to ORF61 in all the seven TGs analysed (donors 1–7; Fig. 2, Supplementary Figs. 5 and 6 and Supplementary Tables 1 and 2). Manual inspection of the VZV sequence read data, combined with de novo transcript reconstruction, revealed five distinct exons (Figs. 2 and 3), two of which encode cleavage factor I-binding motifs (TGTA), while the most 3' exon contained a canonical polyadenylation signal site (AAUAAA) (Fig. 3). We term this the VZV latency-associated transcript (VLT) and subsequently confirmed that VLT nucleotides 40–460 (spanning exons 1–5) are detected as a single transcript using cDNA obtained from the five TGs with the highest VLT expression (Fig. 4a and Supplementary Table 1). Except for ORF63 transcripts detected in six of the seven TGs (donors 2–7), no other VZV transcripts (Fig. 2b and Supplementary Fig. 6) or miRNAs were detected in human TGs.

### Distinct VLT mRNA isoforms expressed during lytic infection.
The VLT locus was also transcribed in lytically VZV-infected ARPE-19 and MeWo cells (Figs. 2b and 4c and Supplementary Fig. 6), enabling rapid amplification of cDNA ends (RACE) to determine 5' and 3' transcript boundaries. Unlike the single VLT isoform detected in latently VZV-infected human TG (Figs. 2 and 4a, b), gene transcription from the VLT locus appears

**Fig. 1** HSV-1 transcriptome profile during lytic and latent infection. Strand-specific mRNA-seq of lytically HSV-1-infected ARPE-19 cells and seven latently HSV-1-infected human trigeminal ganglia (TG) (Supplementary Table 1). **a** Circos plots of the HSV-1 genome [purple band; sense and antisense open reading frames (ORFs) indicated as blue and red blocks, respectively], the latency associated transcripts (LATs) indicated as green blocks, with internal tracks revealing the lytic (left) and latent (right) transcriptomes using unenriched (grey tracks) and HSV-1-enriched (black tracks) libraries. Lytic transcriptomes were profiled using HSV-1-infected ARPE-19 cells. Latent HSV-1 transcriptomes are profiled from seven TGs, with each track depicting a single specimen. Peaks facing outward from the centre indicate reads mapping to the sense strand, while peaks facing inward originate from the antisense strand. The y axis is scaled to maximum read depth per library in all cases. **b** Linear representation of the HSV-1 LAT genomic region (green blocks in **a**), with blue and yellow tracks depicting HSV-1-enriched mRNA-Seq reads originating from the sense and antisense strand, respectively. Unenriched mRNA-Seq tracks for ARPE-19 cells, and TGs 1 and 2, are superimposed and shown in light blue (sense) or yellow (antisense), with overlapping regions in medium-blue and orange, respectively. HSV-1 genome coordinates are shown the HSV-1 reference strain 17 (NC_001806.2); Previously described HSV-1 ORFs within this locus (red boxes), miRNAs (orange blocks), LAT-encoded ORFs (green blocks), LAT-encoded small RNAs (dark red blocks) and LAT (blue boxes) are scaled representatively. . Paired-end read data sets were generated with read lengths of 2 × 34 bp (ARPE-19) or 2 × 76 bp (TG1 and TG2) or 2 × 151 bp (TG3–TG7)

extremely complex during lytic VZV infection of ARPE-19 and MeWo cells (Figs. 2b and 4c, respectively), with additional upstream exons or alternative splicing in exons 3 and 4. Reverse transcriptase–quantitative PCR (RT-qPCR) did not detect any of the three most abundant lytic VLT (hereafter VLT$_{ly}$) isoforms (each VLT$_{ly}$ isoform uses a different upstream exon) in latently VZV-infected TGs (Fig. 4d). Neither additional upstream exons nor alternative splicing of exons 3 and/or 4 were evident in the RNA-Seq data of human TG-derived VZV RNA. Thus a single unique VLT isoform is selectively expressed in latently VZV-infected human TG with short PMI.

**VZV latency is characterized by VLT and ORF63 expression.** To confirm that VLT expression is a general feature of VZV latency, we analysed TG specimens from 18 individuals (donors 1–18; Supplementary Table 1) for the presence of VZV and HSV-1 DNA and transcripts by qPCR and RT-qPCR, respectively. Thirteen TG were co-infected with VZV and HSV-1, while the remaining TG contained only VZV (donors 10 and 12) or HSV-1 (donors 16–18), (Supplementary Table 1). Fourteen of the 15 (93%) VZV nucleic acid-positive (VZV$^{POS}$) TGs expressed VLT and 9 of the 15 (60%) VZV$^{POS}$ TGs co-expressed ORF63 mRNA at lower levels relative to VLT (Fig. 4e, Supplementary Fig. 7a and

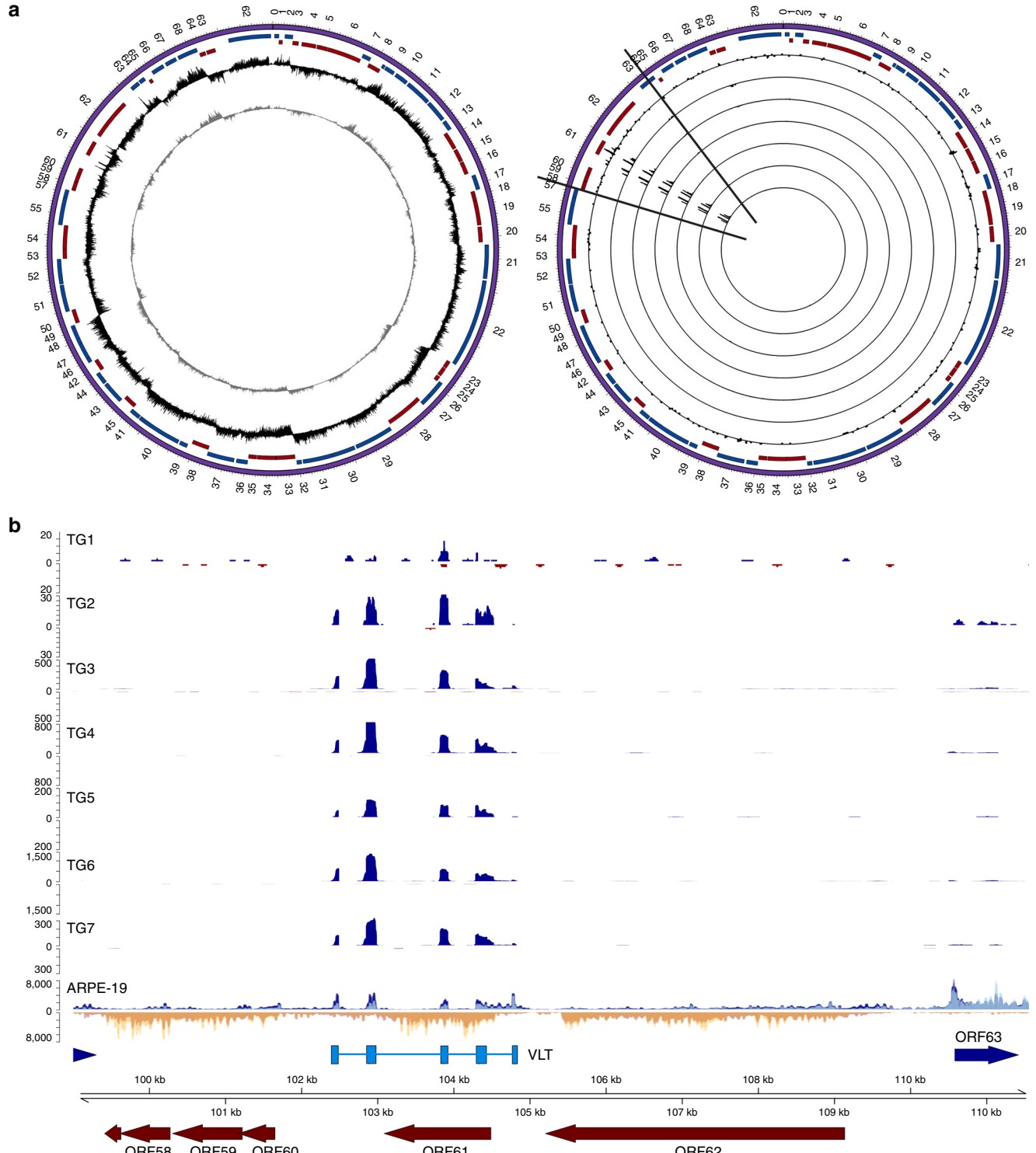

Supplementary Table 1). VLT levels correlated significantly with ORF63 transcript levels (Fig. 4f and Supplementary Fig. 7b) but not with VZV DNA load or PMI excluding the possibility of viral reactivation after death (Supplementary Fig. 7c, d)[18]. Next, we investigated the expression and localization of VLT, and as a control, VZV ORF63 transcript, in latently VZV-infected TGs (n = 12) and two VZV naive human fetal dorsal root ganglia (DRG) by in situ hybridization (ISH). VLT and ORF63 transcripts were localized to both the neuronal nucleus and cytoplasm of distinct neurons in latently VZV-infected TG but not in uninfected DRG (Fig. 4g–j and Supplementary Fig. 8). Significantly more (p = 0.04; paired Student's t-test) TG neurons expressed VLT than ORF63 transcript (Fig. 4g). RNase but not DNase treatment abolished ORF63- and VLT-specific ISH staining (Supplementary Fig. 8), confirming detection of VZV transcripts and not viral genomic DNA. In agreement with previous studies, both RT-qPCR and ISH data (Supplementary Table 1 and Supplementary Fig. 9) revealed a much higher abundance and prevalence of HSV-1 LAT compared to VLT and VZV ORF63 RNA in human TG specimens[31].

**VLT encodes a protein that is expressed in lytic infection**. In silico translation of the VLT isoform expressed in human TG predicted a 136 amino acid protein (pVLT), with a start codon and polyadenylation site/stop codon in exons 2 and 5, respectively (Fig. 3b). A polyclonal pVLT-specific antibody generated against the first 19 N-terminal residues of pVLT (Fig. 3c) confirmed pVLT expression in VLT-transfected ARPE-19 cells (Fig. 5a). In lytically VZV-infected ARPE-19 cells, pVLT is co-expressed by cells expressing ORF62 protein (IE62) and glycoprotein E but not by uninfected cells (Fig. 5b). The kinetic class of $VLT_{ly}$ was determined by RT-qPCR in VZV-infected ARPE-19 cells cultured in the presence or absence of phosphonoformic acid (PFA), a broad-spectrum herpesvirus DNA polymerase inhibitor[32]. Whereas no effect on ORF61 (IE gene) and ORF29 (E gene) transcription was observed, ORF49 (leaky L gene) transcription was markedly reduced in PFA-treated VZV-infected cells (Fig. 5c). PFA blocked $VLT_{ly}$ expression completely (Fig. 5c), demonstrating that $VLT_{ly}$ transcription follows a true late kinetic pattern in vitro. Finally, shingles skin biopsies were assayed for the expression of $VLT_{ly}$ and pVLT. ISH revealed $VLT_{ly}$ expression throughout the affected epidermis and dermis surrounding skin vesicles (Fig. 5d), while IHC analyses of consecutive sections for VZV ORF63 protein (IE63) and pVLT indicated co-expression in the same skin areas but not in healthy control skin (Fig. 5e and Supplementary Fig. 10). Neither pVLT- nor IE63-specific IHC signal was detected in latently VZV-infected human TG sections.

**VLT represses ORF61 expression in co-transfected cells**. Because VLT is antisense to VZV gene ORF61, a major viral transactivator and αHV ICP0 homologue[33], we tested whether VLT affects ORF61 expression. ARPE-19 cells were co-transfected with four plasmids, each containing a single coding VZV sequence (ORFs 61, 62 and 63 and the mature, spliced coding sequence of VLT) under the control of a chicken beta-actin (for ORF61, ORF63 and VLT) or human cytomegalovirus promoter (for ORF62). VLT expression significantly reduced ORF61, but not ORF62 and ORF63 transcript levels, in co-transfected cells (Fig. 6a and Supplementary Fig. 11). Western blot analysis confirmed that VLT diminishes IE61 but not IE62, IE63 and α-tubulin protein abundance in co-transfected cells (Fig. 6b). Mutation of the pVLT start codon within exon 2 (Fig. 3b) from ATG to ATA resulted in loss of pVLT expression in transfected ARPE-19 cells (Fig. 6b and Supplementary Fig. 12) but did not abolish the inhibitory effect of VLT on ORF61 transcription and protein (IE61) expression in co-transfected cells (Fig. 6b). The data implicate VLT, but not pVLT, in the selective repression of VZV ORF61 gene expression.

## Discussion

We have identified a unique spliced VZV transcript, VLT, which is consistently expressed in latently VZV-infected human TG neurons. The VLT locus, including splice donor/acceptor sites and pVLT coding sequence, is highly conserved between wild-type and vaccine strains of VZV (Supplementary Table 4). A feature shared by well-characterized latency transcripts of other αHVs, notably HSV-1 and BHV-1, is their ability to encode repressive miRNAs[13,14]. However, in keeping with previous analyses of human TG[34], we found no evidence of miRNAs encoded within VLT or the wider viral transcriptome. In contrast to other αHVs, latent VZV also transcribes the lytic ORF63 gene in a subset of latently infected TGs, suggesting an additional role for this viral gene in latency or early reactivation. Using a highly sensitive enriched RNA-Seq method, we found no evidence for the expression of other VZV genes during latent infection. Nor did we find evidence that HSV-1 latency in human TG involves the expression of viral transcripts other than the 8.3 kb full-length and 1.5/2.0 kb spliced HSV-1 LATs and associated miRNAs. Unlike HSV-1 LAT, VLT encodes a protein (pVLT) whose expression was strictly dependent on de novo virus DNA replication (true late kinetics) in lytically VZV-infected cells in vitro and in shingles skin lesions but was undetectable by IHC during latency.

The functions of VLT during latency and the $VLT_{ly}$ isoforms expressed during lytic VZV infection remain to be elucidated. Deletion of VZV ORF61, which would disrupt VLT, does not affect establishment of quiescent neuronal infection in the cotton tailed rat[35], while a SVV ORF61 deletion mutant, which would disrupt the putative SVV LAT, could still establish latency in its natural host[36]. We have shown that in co-transfected cells VLT specifically suppresses the expression of VZV ORF61, an αHV ICP0 homologue and a promiscuous transactivator of lytic viral promotors[37]. The recent development of tractable in vitro cell

**Fig. 2** VZV transcriptome profile during lytic and latent infection. Strand-specific mRNA-seq of lytically VZV-infected ARPE-19 cells and seven latently VZV-infected human trigeminal ganglia (TG) (Supplementary Table 1). **a** Circos plots of the VZV genome [purple band; sense and antisense open reading frames (ORFs) indicated as blue and red blocks, respectively], with internal tracks revealing the lytic (left) and latent (right) transcriptomes using unenriched (grey track, left panel) and VZV-enriched (black tracks, left and right panels) libraries. Right panel: latent VZV transcriptome of seven TG, with each track depicting a single specimen. Peaks facing outward from the centre indicate reads mapping to the sense strand, while peaks facing inward originate from the antisense strand. The y axis is scaled to maximum read depth per library in all cases. **b** Linear representation of the varicella latency-associated transcript (VLT) genomic region (black lines in **a**), with blue and yellow tracks depicting VZV-enriched mRNA-Seq reads originating from the sense and antisense strands, respectively. Unenriched mRNA-Seq tracks for ARPE-19 cells, and TGs 1 and 2, are superimposed and shown in light blue (sense) or yellow (antisense), with overlapping regions in medium-blue and orange, respectively. No VZV-mapping reads were obtained from unenriched sequence data sets generated from TGs 1 and 2. VZV genome coordinates are shown the VZV reference strain Dumas (NC_001348.1); blue and red arrows indicate previously described VZV ORFs, and light blue boxes indicate the five VLT exons. Paired-end read data sets were generated with read lengths of 2 × 34 bp (ARPE-19) or 2 × 76 bp (TG1 and TG2) or 2 × 151 bp (TG3–TG7)

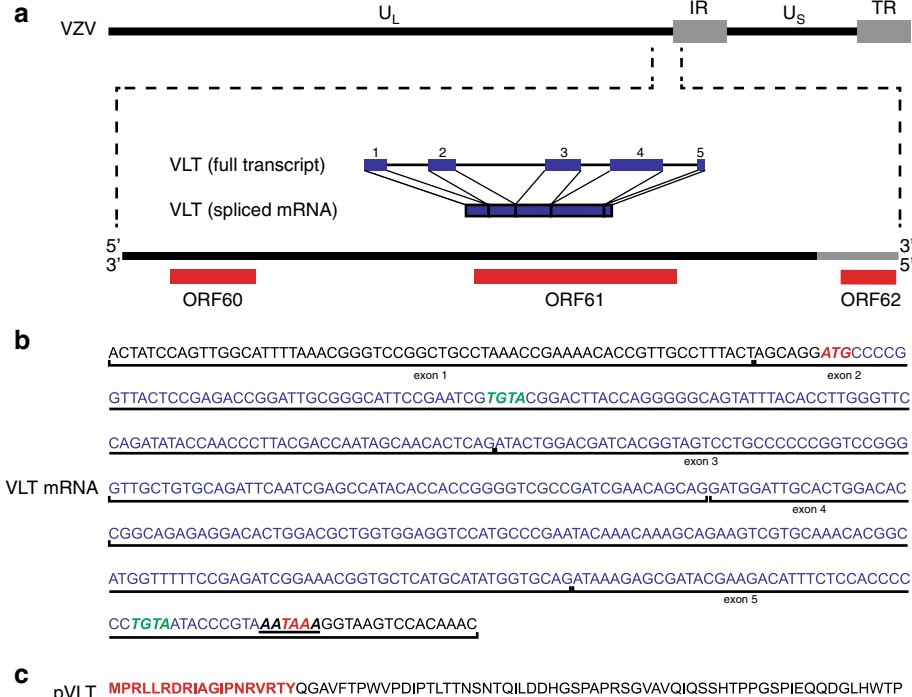

**Fig. 3** The genomic locus encoding the VZV latency-associated transcript (VLT). **a** Schematic diagram showing the location and structure of the five VLT exons (blue blocks) and introns (blue lines) within the genomic region 101,000–106,000 (coordinates refer to VZV reference strain Dumas; NC_001348.1) (Supplementary Table 3). **b** The VLT mRNA sequence including the 5′ untranslated region; start and stop codons are highlighted in red italic, while the cleavage factor I (CFI)-binding motifs are highlighted in green italic and the canonical polyadenylation signal site (AATAAA) is underlined. Location and boundaries of VLT exons are indicated by vertical black lines. **c** The fully translated VLT protein (pVLT), with the sequence of peptide (red) used to produce rabbit polyclonal anti-pVLT antibody

models[25],[26] and human fetal DRG organotypic cultures[38],[39] of VZV latency and reactivation, although not completely representative of the latent TG state, also provides exciting opportunities to explore further VLT function[40].

With the availability of effective vaccines, VZV disease is much reduced in those geographic regions where coverage is high[41]–[43]. However, naturally infected individuals and vaccine recipients still harbour latent virus (wild type and/or vaccine strains) that can reactivate to cause disease[44]. The identification of the VLT, 35 years after VZV latency in human sensory ganglia was first described[45], provides a significant advance in ongoing efforts to investigate and understand the mechanisms by which VZV establishes, maintains and reactivates from latency.

## Methods

**Human clinical specimens**. Human TG were obtained at 6:01 h ± 1:47 h (average ± SD) after death (Supplementary Table 1). The brain samples and/or bio samples were obtained from The Netherlands Brain Bank (NBB), Netherlands Institute for Neuroscience, Amsterdam (open access: www.brainbank.nl). All Material has been collected from donors for or from whom a written informed consent for a brain autopsy and the use of the material and clinical information for research purposes had been obtained by the NBB. All study procedures were performed in compliance with relevant Dutch laws and institutional guidelines, approved by the local ethical committee (VU University Medical Center, Amsterdam, project number 2009/148) and was performed in accordance with the ethical standards of the Declaration of Helsinki. Majority of TG donors had a neurologic disease history affecting the central nervous system (mainly Alzheimer's and Parkinson's disease) and cause of death was not related to herpesvirus infections. TG biopsies were either formalin-fixed and paraffin-embedded (FFPE) or snap-frozen in liquid nitrogen and stored at −80 °C. Human fetal DRG were obtained from the Academic Medical Centre Amsterdam (the Netherlands) according to relevant Dutch laws and approved by the institutional ethical committee (Erasmus MC, Rotterdam, MEC-2017-009). FFPE punch biopsies of one healthy control subject and five herpes zoster skin lesions were obtained for diagnostic purposes. Zoster biopsies were confirmed VZV DNA positive by virus-specific real-time PCR (qPCR). According to the institutional 'Opt-Out' system (Erasmus MC, Rotterdam, the

Netherlands), which is defined by the National 'Code of Good Conduct' [Dutch: Code Goed Gebruik, May 2011], the surplus human herpes zoster FFPE tissues were available for the current study.

**Cells and viruses**. Human retinal pigmented epithelium ARPE-19 cells [American Type Culture Collection (ATCC) CRL-2302] were maintained in 1:1 Dulbecco's modified Eagle's medium (DMEM; Lonza or Nissui)–Ham's F12 (Lonza or Sigma-Aldrich) medium supplemented with 8% heat-inactivated foetal bovine serum (FBS; Lonza or Sigma-Aldrich) and 0.6 mg/mL L-sodium glutamate (Lonza or Nacalai). The human MeWo melanoma cell [ATCC HTB-65] was maintained in DMEM supplemented with 8% FBS and 0.6 mg/mL L-sodium glutamate. Culture of VZV pOka strain and isolation of cell-free virus have been described previously[46]. Low passage clinical isolates VZV EMC-1 and HSV-1 strain F (ATCC VR-733) were cultured on ARPE-19 cells as described[47]–[49]. All cell cultures and virus infections were performed in a humidified $CO_2$ incubator at 37 °C.

**Nucleic acid extraction from human TG**. Approximately one-fifth of snap-frozen human TG was mechanically dispersed and used for DNA isolation, while four-fifths of the same specimen was used for RNA isolation. DNA extraction was performed using the QIAamp DNA Kit (Qiagen) according to the manufacturer's instructions. For RNA isolation, samples were homogenized in TRIzol (Invitrogen), mixed vigorously with 200 μL chloroform and centrifuged for 15 min at 12,000×g at 4 °C. RNA was isolated from the aqueous phase using the RNeasy Mini Kit, including on-column DNase I treatment (Qiagen). DNA and RNA concentration and integrity were analysed using a Qubit Flourometer (ThermoFisher).

**RNA extraction from lytic VZV and HSV-1 infections**. ARPE-19 cells were infected with VZV strain EMC-1 by co-cultivation VZV-infected or uninfected cells at a 1:8 cell ratio and harvested in 1 mL TRIzol at 72 h post-infection (hpi). Alternatively, semi-confluent ARPE-19 cell layers (75 cm² flask) were infected with HSV-1 strain F at a multiplicity of infection of 1 and harvested in 1 mL TRIzol at 24 hpi. RNA isolated as described above was subjected to additional DNase treatment using the TURBO DNA-free Kit according to the manufacturer's instructions (Ambion).

**RNA-Seq library preparation and sequencing**. Four micrograms of total RNA was used as input for the SureSelectXT RNA Target Enrichment protocol (Agilent Technologies G9691 version D0). Here each sample was either enriched for

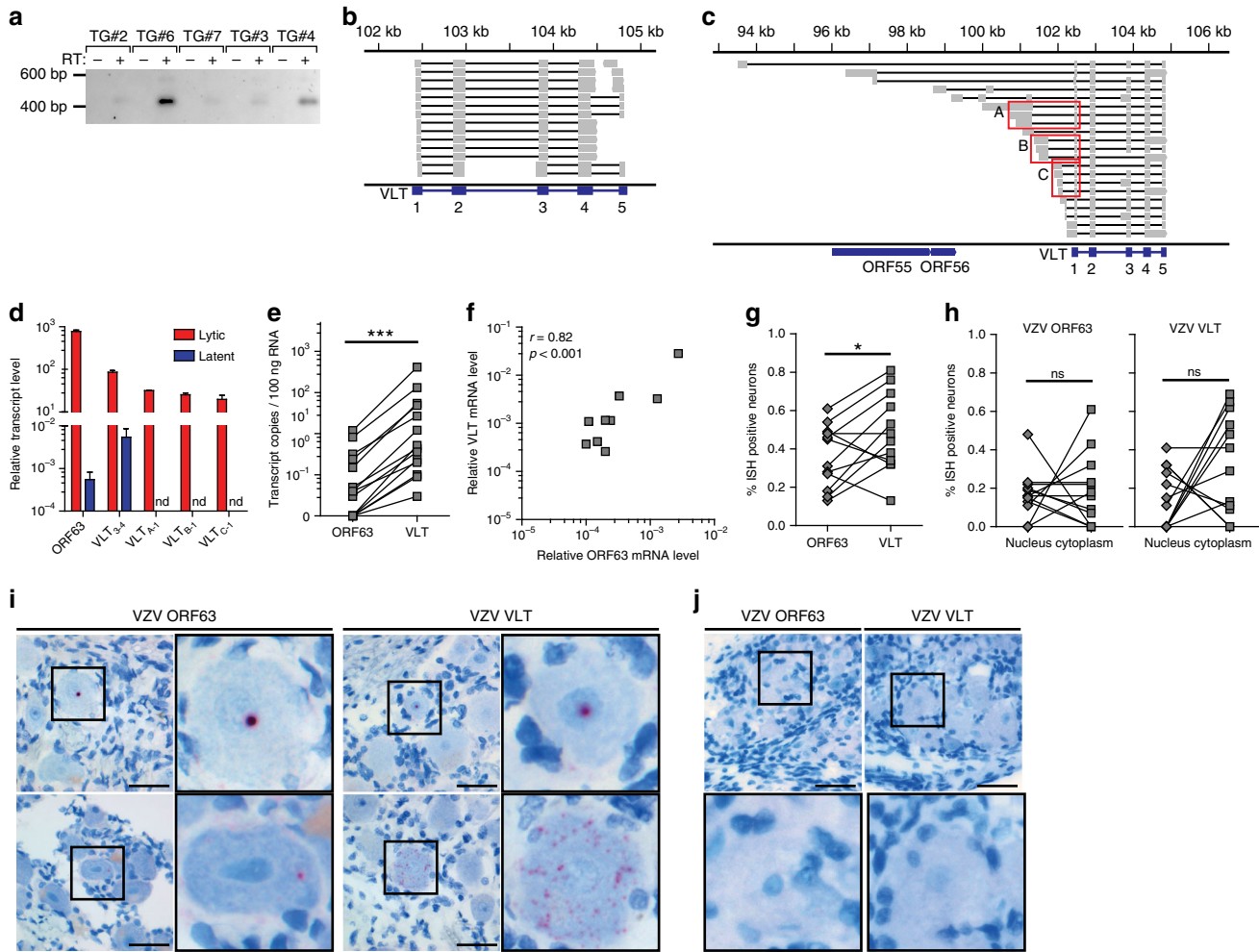

**Fig. 4** Prevalence of VLT and ORF63 transcript in human TG. **a** PCR amplification of cDNA (n = 5 TGs, Supplementary Table 1), synthesized in the presence (+) or absence (−) of reverse transcriptase. Sanger sequencing of all five purified PCR products yielded identical sequences corresponding to VLT (Fig. 3b). **b** Integrative Genomics Viewer (IGV) screenshot showing representative RNA-Seq data from a single TG. Paired-end reads are shown as fragments mapped across the VLT locus. Grey boxes (exons) connected by black lines (introns) indicate individual read pairs with distinct read-pairs separated by white space. Each fragment spans between 2 and 5 exons and there was no evidence of additional upstream exons. **c** In lytically VZV-infected MeWo cells, Sanger sequencing of amplicons generated through rapid amplification of cDNA ends revealed multiple VLT$_{ly}$ isoforms, visualized using IGV. Sequencing of 29 clones identified the three most abundant VLT$_{ly}$ isoform groups (red boxes) A (35%), B (19%) and C (15%). Note that the specific VLT isoform observed in latently infected TG was not observed among VLT$_{ly}$ isoforms. **d** Quantification of ORF63 transcript and VLT isoforms in lytically VZV-infected ARPE-19 cells (red; n = 3) and latently VZV-infected TG (blue; n = 19 TG), using primers/probes spanning VLT splice junctions between exons 3→4, A→1, B→1 and C→1 (Fig. 4c and Supplementary Table 5). Data represent mean (±SEM) relative transcript levels normalized to β-actin RNA. nd not detected. **e** Levels of paired ORF63 transcript and VLT (primers/probe spanning exon 2→3) in the same VZV$^{POS}$ TG (VZV$^{POS}$; n = 15) determined by RT-qPCR. ***p < 0.001; Wilcoxon signed rank test. **f** Correlations (Spearman) between relative ORF63 transcript and VLT levels in VZV$^{POS}$ TG (n = 15), as determined by RT-qPCR. **g–j** In situ hybridization (ISH; red signal) analysis of ORF63 RNA and VLT in latently VZV-infected TG (n = 12). **g** Frequency of neurons positive for ORF63 RNA and VLT in consecutive TG sections of the same donor. *p < 0.05; paired Student's t-test. **h** Nuclear and cytoplasmic ORF63 RNA and VLT expression in consecutive TG sections from individual donors. ns not significant; paired Student's t-test. **i, j** Representative ISH images of VZV$^{POS}$ TG sections (**i**) and two VZV naive human fetal dorsal root ganglia (**j**). Nuclei were stained with haematoxylin. Magnification: ×400 (with 3× digital zoom for insets). Bars = 50 μm

polyadenylated mRNA (captured by oligo-dT beads, as described by the SureSelect XT protocol) or underwent rRNA-depletion using a NEBNext® rRNA Depletion Kit [Human/Mouse/Rat] (New England Biolabs) according to the manufacturer's instructions. Subsequent to either procedure, captured/remaining RNAs were transcribed to produce cDNA. Fragmentation, cDNA second-strand synthesis, end repair, A-tailing and adapter ligation were performed as described in the enrichment protocol. Hybridization was performed using a modified strategy[50,51] that incorporated custom-designed SureSelect RNA bait sets for both VZV and HSV-1 in the same reaction at reduced concentration (1:10). Bait sets for VZV (Supplementary Data 1) and HSV-1 (Supplementary Data 2) were designed at 12× tiling (i.e., each base position in the genome was covered by 12 distinct 120-mer bait sequences) using custom in-house scripts. Hybridization for 24 h at 65 °C was followed by post-capture washing and optimized PCR-based library indexing (12 cycles for RNA obtained from lytically infected cultures, 18 cycles for RNA obtained from TGs). Libraries generated from VZV- and HSV-1-infected ARPE-19 cells were multiplexed and sequenced using a 75 cycle V2 high-output kit. Subsequently, three TG were multiplexed sequenced using a 150 cycle V2 mid-output kit followed by a further four TG multiplexed and sequenced using a 300 cycle V2 high-output kit. The decision to use increasing read lengths was informed by the initial discovery of VLT and the desire to better characterize intron splicing.

**Transcriptome mapping and de novo transcript reconstruction.** Sequence run data were de-multiplexed using bcl2fastq2 v2.17 under stringent conditions (--barcode mismatches 0) and yielded between ~31,000,000 and 107,700,000 paired-end reads per sample. Individual sequence data sets were trimmed using the TrimGalore software (http://www.bioinformatics.babraham.ac.uk/projects/

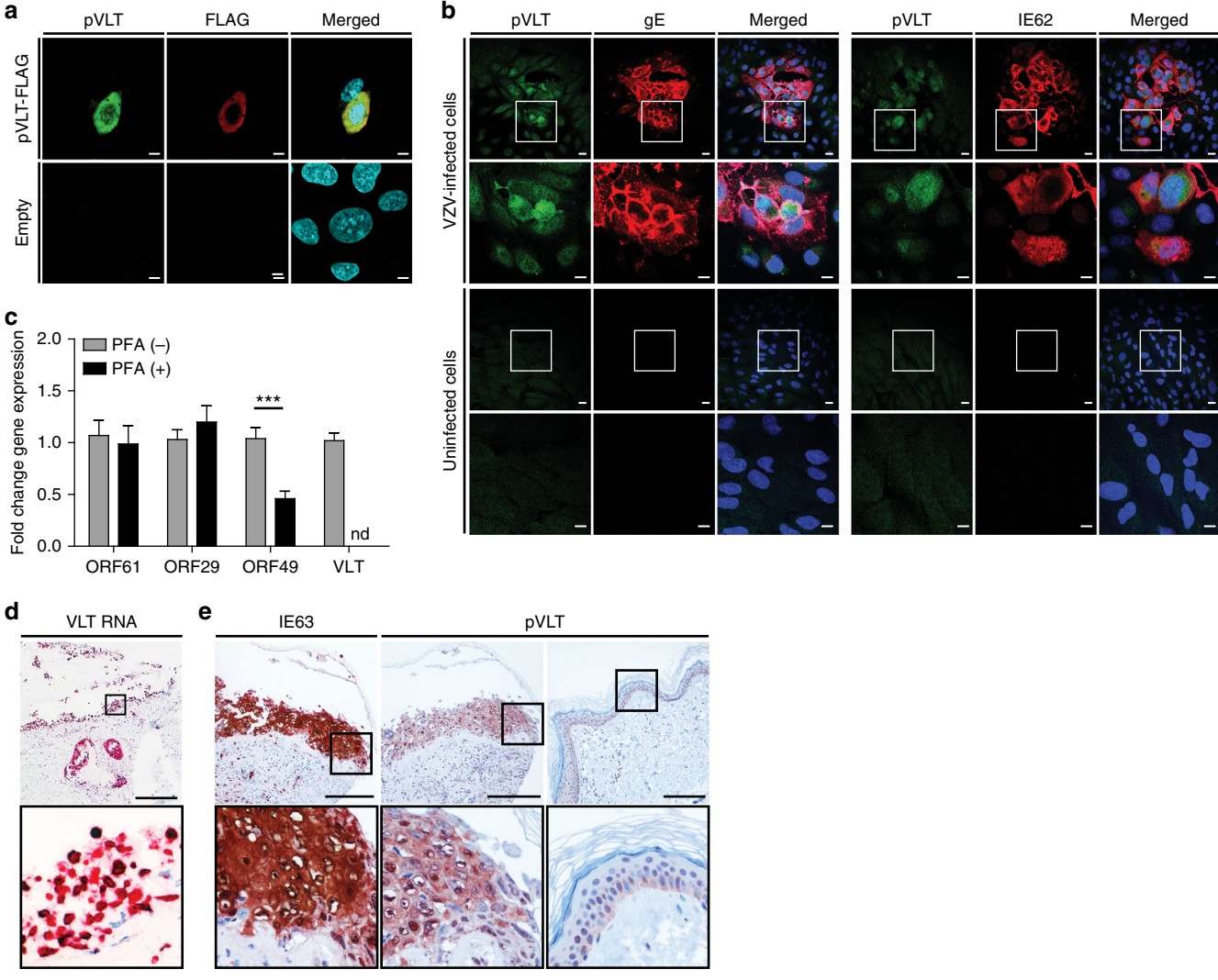

**Fig. 5** Expression of VLT protein in vitro and in situ. **a** Representative confocal microscopic image of ARPE-19 cells at 48 h post-transfection with C-terminal FLAG-tagged VLT (pVLT-FLAG) expression plasmid or empty control plasmid (empty). pVLT-FLAG was detected with antibodies directed to FLAG (red) and VLT protein (pVLT; green). Magnification: ×1000 and ×2 digital zoom. Bars = 5 μm. **b** Representative confocal microscopic images of uninfected and VZV strain EMC1-infected ARPE-19 cells at 2 days post-infection, stained for both pVLT (green) and VZV glycoprotein E (gE; red) (left panel) or pVLT (green) combined with VZV ORF62 protein (IE62; red) (right panel). Magnification: ×200, with area indicated by the white box shown at ×600. Bars = 20 μm (×200) and 10 μm (×600). In **a**, **b**, nuclei were stained with Hoechst 33342 (cyan) and images are representative of results from four independent experiments. **c** RT-qPCR quantitation of VZV ORF61, ORF29 and ORF49 transcript and VLT levels in VZV pOka-infected ARPE-19 cells cultured with (PFA +) and without phosphonoformic acid (PFA−) for 24 h. Data represent mean (±SEM) fold-change in gene expression, using the respective 'PFA−' value as a calibrator, from four independent experiments. ***$p < 0.001$; Wilcoxon signed rank test. **d** Detection of VLT (red) in a human herpes zoster (HZ) skin lesion by in situ hybridization. Magnification: ×100. Inset: ×400 and 2× digital zoom. Bar = 200 μm. **e** Consecutive HZ skin sections stained immunohistochemically for ORF63 protein (IE63; brown) and pVLT (brown) (left and middle panels) and sections from unaffected control skin stained for pVLT (right panel). Magnification: ×200. Inset: ×400. Bars = 100 μm. Images in **d**, **e** are representative of five HZ skin biopsies stained

trim_galore/) to remove low-quality 3' ends and mapped to the human genome (hg19) using the BBMap software (http://sourceforge.net/projects/bbmap/) with default parameters. Unmapped read pairs were subsequently aligned against VZV Dumas (NC_001348) and HSV-1 strain 17 (NC_001806) reference genomes deposited at Genbank using BBMap allowing for properly paired reads only to be carried forward in the analysis. Extremely high duplication levels (>0.99) were observed in all enriched libraries generated from TG RNA, an expected feature of enrichment strategies against ultra-low abundance transcripts. Note, however, that we are not stating that VLT is a low-abundance transcript within a single cell but rather that VZV latently infected cells are relatively scarce, thus diluting the apparent expression level of VLT. To mitigate this, duplicate reads were removed using the Picard Tools MarkDuplicates software (http://broadinstitute.github.io/picard). Resulting assemblies were visualized using a combination of Circos[52], Artemis[53], Tablet[54], SeqMonk (http://www.bioinformatics.babraham.ac.uk/projects/seqmonk/) software and custom R scripts making use of the Rsamtools and Gviz software packages.

To confirm that enrichment for viral nucleic acids did not bias relative levels of viral gene transcription, transcripts per million counts were generated using FeatureCounts (subread package[55,56]) and visualized using scatter plots. High correlation ($R^2$ 0.9252–0.9678) between enriched and unenriched transcriptomes were observed for both HSV-1 ($n = 2$ biological replicates) and VZV ($n = 1$), as shown in Supplementary Fig. 1.

For de novo transcript reconstruction, VZV-specific mapping read pairs for each TG were extracted from BAM files and converted to raw fastq format for input into Trinity[57]. Trinity enables transcript reconstruction and, while limited in this case by the scarcity of VZV reads, was able to merge overlapping reads to produce transcript isoforms that, when mapped to the VZV reference genome using BBmap, spanned the four major introns observed in the VLT.

**MicroRNA sequencing (miRNASeq) and qPCR profiling.** miRNASeq libraries were prepared from 1 μg of total RNA, isolated from human TG and lytically VZV- and HSV-1-infected ARPE-19 cells using the NEBNext® small RNA Library Prep

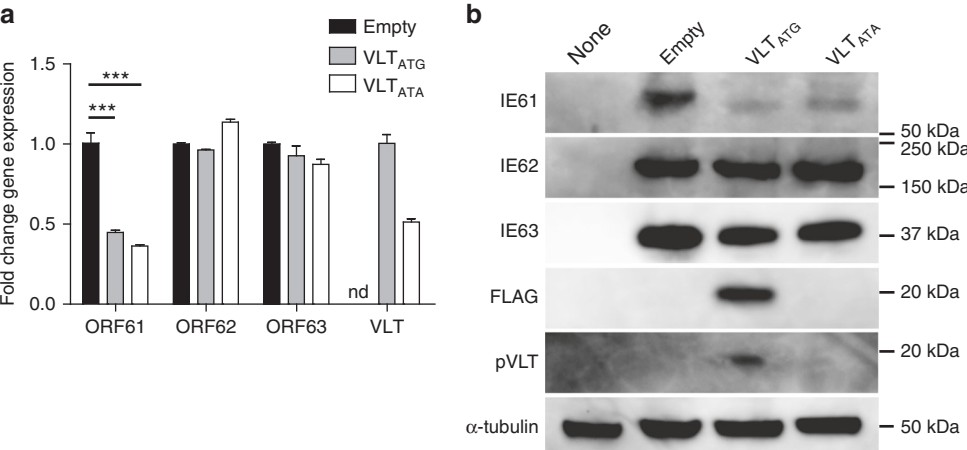

**Fig. 6** Selective repression of VZV ORF61 gene expression by VLT in co-transfected cells. ARPE-19 cells were transfected with plasmids encoding FLAG-tagged VLT (VLT$_{ATG}$), mutated VLT in which the ATG start codon was replaced by ATA sequence (VLT$_{ATA}$) or empty control plasmid (empty) in combination with plasmids encoding ORF61, ORF62 and ORF63. **a** Analysis of VZV ORF61, ORF62 and ORF63 and human β-actin transcript by RT-qPCR. Data represent mean (±SEM) fold-change in gene expression using the empty vector to calibrate ORF61, ORF62 and ORF63 or using VLT$_{ATA}$ to calibrate VLT$_{ATG}$. Data are from two independent experiments performed in duplicate. nd not detected. ***$p < 0.001$; one-way ANOVA with Bonferroni's correction for multiple comparisons. **b** Western blot analysis using antibodies directed to proteins encoded by VZV ORF61 (IE61), ORF62 (IE62), ORF63 (IE63), VLT (pVLT) and FLAG-tagged pVLT and α-tubulin. None, untransfected ARPE-19 cells. Images are representative of four independent experiments

Set for Illumina® according to the manufacturer's instructions. Libraries underwent 75 bp single-end sequencing using an Illumina NextSeq prior to demultiplexing (as outlined above). Sequence reads were adaptor-trimmed using TrimGalore and size selected using BBduk (provided with BBmap) so that only sequence reads between 17 and 26 bases in length were retained. These were mapped against VZV strain Dumas and the HSV-1 strain 17 reference genomes as miRDB using ShortStack (--dicermin 18 --mincov 1 --mismatches 2 --foldsize 200). No putative VZV miRNAs could be detected in either latently infected TGs or lytically infected ARPE-19 cells. By contrast, canonical HSV-1 miRNAs could be detected in both latently infected TGs and lytically infected ARPE-19 cells at high abundance.

**cDNA synthesis and qPCR.** cDNA synthesis was performed as described[18] using 2.0–11.7 μg [6.7 μg ± 0.64 (average ± SEM)] of TG-derived total RNA or 5 μg of cell-culture-derived DNase-free RNA and Superscript III reverse transcriptase (Invitrogen) with oligo(dT)12-18 primers (Invitrogen). Taqman qPCR was performed in triplicate on DNA and cDNA using Taqman 2× PCR Universal Master Mix (Applied Biosystems) with primer/probe pairs specific for VZV ORF62, ORF63 and the VLT exon 2→3, 3→4, A→1, B→1 and C→1 junctions and HSV-1 DNA[58] and LAT, (all from Eurogentec), β-actin (Applied Biosystems) and the human single copy gene hydroxymethylbilane synthase (Supplementary Table 5). Commercially quantified VZV and HSV-1 DNA stocks (Advanced Biotechnologies) and plasmids encoding VZV DNA amplicons (ORF63 and VLT; described in 'Plasmid construction' section below) were used to standardize qPCR reactions and used as positive control in each qPCR. RT-qPCR data were presented as absolute transcript copy number per 100 ng RNA (Fig. 4e and Supplementary Figs. 7a and 9a), relative transcript levels defined as $2^{-(Ct\text{-value VZV gene}-Ct\text{-value β-actin})}$ (Fig. 4d, f and Supplementary Figs. 7b–d and 11) and fold change in gene expression using the $2^{-\Delta\Delta Ct}$ method[59] in which transcript levels were normalized to β-actin and a reference sample (Figs. 5c and 6a).

**Sequence analysis of lytic and latent VLT isoforms.** PCR was performed on cDNA obtained from five human TG donors [TGs 2, 3, 4, 6 and 7, which expressed the highest VLT levels (Supplementary Table 1)] using Pfu Ultra II Fusion HS DNA Polymerase (Stratagene) and primers ORF61-13 and VLT_exon5-Rv (Supplementary Table 5). 5'RACE and 3'RACE were performed using the SMARTer™ RACE cDNA Amplification Kit (Clontech) according to the manufacturer's instructions using total RNA extracted from VZV pOka-infected MeWo cells. PCR was performed using KOD-Plus-Ver.2 DNA Polymerase (Toyobo Life Science) templated with 5'RACE ready cDNA using Universal Primer A Mix and ORF61R626AS or pOka104778R or 3'RACE ready cDNA using Universal Primer A Mix and ORF61R626AS (Supplementary Table 5). Alternatively, 5'RACE was performed using the FirstChoice™ RLM-RACE Kit (Ambion) according to the manufacturer's instructions, using 10 μg of total RNA obtained from VZV EMC-1-infected ARPE-19 cells. Nested PCR was performed using Pfu Ultra II Fusion HS

DNA Polymerase (Stratagene) and 40 cycles of first round amplification using 5'RACEout_Fw–VLT2-3_Rv primers, followed by 40 cycles of second round amplification using 5'RACEinn_Fw–VLT2_Rv primers (Supplementary Table 5). PCR products were cloned into the pCR™4-TOPO® TA vector (Thermo Fisher Scientific) after addition of 3'A overhang using AmpliTaq Gold DNA polymerase (Thermo Fisher Scientific) or Takara Ex Taq® (Takara Bio) and used to transform One Shot® TOP10 competent *Escherichia coli*. Plasmid DNA was extracted using the QIAprep Spin Miniprep Kit (Qiagen), amplified by PCR using M13Fw and M13Rv primers (Supplementary Table 5) and Pfu Ultra II Fusion HS DNA Polymerase (Stratagene). Plasmid DNAs or purified PCR products were sequenced on the ABI Prism 3130 XL Genetic Analyzer using the BigDye v3.1 Cycle Sequencing Kit (both Applied Biosciences) and M13Fw and M13Rv primers. Resultant FASTA sequences were aligned the VZV reference genome Dumas using BBmap, as outlined above. Resultant assemblies were inspected alongside RNA-Seq data using IGV[60].

**In situ hybridization.** To select for VZV and HSV-1 DNA-positive TG tissue areas, DNA was isolated from three consecutive 5 μm FFPE tissue sections using the QIAamp DNA FFPE Tissue Kit and analysed by the respective virus-specific qPCR assays. Subsequently, viral DNA$^{POS}$ FFPE TG specimens, human fetal DRG (negative control) and human zoster skin biopsies were analysed by in situ hybridization (ISH) using the RNAScope 2.0 Red Kit (Advanced Cell Diagnostics) according to the manufacturer's instructions. In brief, deparaffinized 5 μm tissue sections were incubated with probes designed to cover VZV ORF63 and VLT exons 2–5. The probes for the human transcript POLR2A and ubiquitin C were used as positive controls and probes specific for the bacterial transcript DAPB were used as negative controls. All probes were designed and produced by Advanced Cell Diagnostics. FastRed was used as substrate to visualize the ISH signal and stained slides were counterstained with haematoxylin and mounted in Ecomount (Biocare Medical). In some experiments, TG sections were incubated with DNase I (Qiagen) or RNase [Ribonuclease A (25 μg/mL)+T1 (25 units/mL; both Thermo Fisher Scientific) diluted in 1×TBS-t], after pre-treatment step #3 for 1 h at 40 °C. To determine the ratio of VZV and HSV-1 transcript expressing TG neurons, slides were scanned using the Nanozoomer 2.0 HT (Hamamatsu) and scored in Adobe Photoshop CS6 (Adobe). Twelve TG from distinct donors were analysed for VZV ORF63 transcript and VLT and 10 TG for HSV-1 LAT, with on average 664 neurons/section (range: 420–1561) and 1–2 sections per TG. Two herpes zoster skin biopsies from distinct donors were analysed with 2–3 sections per donor for each staining.

**Determination of kinetic class of VZV transcripts.** ARPE-19 cells ($2 \times 10^5$ cells) were infected with pOka VZV cell-free virus ($10^3$ plaque-forming units) with and without PFA (200 μg/mL) for 24 h at 37 °C. Total RNA was isolated from cells using NucleoSpin RNA in combination with the NucleoSpin RNA/DNA buffer set

(Macherey-Nagel) according to the manufacturer's instructions. Binding DNA was first eliminated from the column in 100 μL DNA elution buffer, the column was treated with recombinant DNase I (5 units/100 μL; Roche Diagnostics) for 1 h at 37 °C and finally RNA was eluted in 60 μL nuclease-free water. cDNA was synthesized with 10.4 μL of total RNA and anchored oligo(dT)$_{18}$ primer in a 20 μL reaction using the Transcriptor High Fidelity cDNA Synthesis Kit (Roche Diagnostics). qPCR following a dissociation curve analysis was performed as described previously[25] using SYBR Select Master Mix in a StepOnePlus Real-time PCR system (Thermo Fisher Scientific). The primer sets for β-actin and VZV ORF61 and ORF49 genes were described previously[25] and primers for VZV ORF29 (ORF29F2381 and ORF29R2440) and VLT (VLTexon1F and VLTexon2R) are presented in Supplementary Table 5.

**Plasmid construction.** The VLT coding sequence (102,468–104,818, excluding introns indicated in Supplementary Table 3) and ORF61 and ORF63 coding sequences were amplified by PCR of cDNA prepared from pOka-infected MeWo cells showing >80% cytopathic effect using the primer sets VLTup22ecoF and VLT-FLAGxhoR, ORF61up20ecoF and ORF61SalR or NPup20ecoF and ORF63xhoR, respectively (Supplementary Table 5). Products were digested with EcoRI and XhoI (VLT and ORF63) or EcoRI and SalI (ORF61) restriction enzymes and subsequently cloned into pCAGGS-MCS-puro (CAG.Empty) via EcoRI and XhoI sites. The resulting VLT-, ORF61- and ORF63-expressing plasmids were named as follows: CAG.VLT-FLAG, CAG.ORF61, and CAG. ORF63, respectively. The CAG.VLT(ATA)-FLAG plasmid, in which the ATG start codon of VLT ORF was mutated to ATA to prevent pVLT expression, was generated using primer VLT-G3A with a QuickChange Lightning Multi Site-Directed Mutagenesis Kit (Agilent Technologies) according to the manufacturer's recommendation based on the CAG.VLT-FLAG. All primers used to construct VZV gene expression plasmids are listed in Supplementary Table 5. The pCAGGS plasmid[61] was generous gift from Dr. Jun-ichi Miyazaki (Osaka University, Japan). The pcDNA.ORF62 was a generous gift from Dr. Yasuyuki Gomi (Research Foundation for Microbial Diseases, Osaka University).

**Generation of rabbit anti-pVLT and -IE63 antibodies.** Anti-VLT protein (pVLT) antibody was generated by Sigma-Aldrich by immunizing a rabbit with a synthetic peptide encoding the first 19 amino acids of pVLT (MPRLLRDRIAGIPNRVRTY; Fig. 3c). The antibody was purified using pVLT peptide-conjugated NHS (N-hydroxysuccinimide)-activated sepharose (GE Healthcare Life Sciences). For probing of ORF63 (NP_040185.1), an anti-IE63 antibody was also generated by Sigma-Aldrich by immunizing a rabbit with GST-IE63 protein as described for anti-IE61 antibody[62]. Briefly, GST-IE63 protein was expressed in and purified from E. coli BL21 transformed with pGEX-IE63, in which the entire DNA fragment except the first ATG (i.e., ORF63 nucleotide positions 4–837) was cloned into pGEX6P-1 bacterial expression vector (GE Healthcare). The anti-IE63 antibody was purified using GST-conjugated NHS-activated sepharose for depleting anti-GST antibody and GST-IE63-conjugated NHS-activated sepharose.

**Immunofluorescent staining and confocal microscopy of cells.** The following primary mouse monoclonal antibodies directed to the indicated proteins were used: VZV IE62 (1:100 dilution; generous gift from Dr. Jürgen Haas (University of Edinburgh, UK) and Dr. Stipan Jonjić (Faculty of Medicine, Rijeka, Croatia))[63], VZV glycoprotein E (1:200 dilution; MAB8612, Millipore) and DYKDDDDK tag (FLAG tag) (1:100 dilution; Clone 1E6, WAKO). Alexa Fluor 488- and Alexa Fluor 594-conjugated goat-anti-rabbit and -anti-mouse IgG (1:250 dilution; Thermo Fisher Scientific) were used as secondary polyclonal antibodies, respectively. Hoechst 33342 (Sigma-Aldrich) was used for nuclear staining. Confocal microscopic analysis were performed as previously described[62,64].

**IHC and immunofluorescence on skin biopsies.** Deparaffinized and rehydrated 5 μm FFPE sections of human herpes zoster skin lesions and healthy control skin were subjected to heat-induced antigen retrieval with citrate buffer (pH = 6.0), blocked and incubated with mouse anti-VZV IE63 (1:1500 dilution; kindly provided by Dr. Sadzot-Delvaux; Liege, Belgium)[65], rabbit anti-pVLT (1:100 dilution) or isotype control antibodies overnight at 4 °C. Sections were subsequently incubated with biotinylated secondary goat-anti-rabbit immunoglobulin (Ig) or goat-anti-mouse Ig and streptavidin-conjugated horseradish peroxidase (all from Dako) for 1 h at room temperature. Signal was visualized using 3-amino-9-ethylcarbazole and counterstained with haematoxylin (Sigma-Aldrich). For immunofluorescent staining, Alexa Fluor 488- and Alexa Fluor 594-conjugated goat-anti-mouse and goat-anti-rabbit antibodies (all 1:250 dilution) were used, and sections were mounted with Prolong Diamond antifade mounting medium with 4,6-diamidino-2-phenylindole. Confocal microscopic analysis was performed as described[64].

**Plasmid co-transfection, RT-qPCR and immunoblotting.** Plasmid co-transfection was performed using PEImax (molecular weight 40,000) (Polysciences, Inc.). High potency linear polyethyleneimine was dissolved in water (1 mg/mL), adjusted to pH = 7 with NaOH, filtered through an 0.22 μm filter and stored in aliquots at −20 °C until use. CAG.Empty, CAG.VLT-FLAG or CAG.

VLT(ATA)-FLAG (2 μg) with CAG.ORF61, pcDNA.ORF62 and CAG.ORF63 (1 μg) were diluted in knockout DMEM/F12 media (Thermo Fisher Scientific) (50 μL) and PEImax solution (6 μL) were diluted in knockout DMEM/F12 media (50 μL), then both diluent were mixed, left at room temperature for 10–15 min to form polyplexes and transfected into ARPE-19 cell (1 × 10$^5$ cells/well in an 12-well plate). Culture medium was changed at 16 h post-transfection and cultured for another 48 h. Cells were harvested and aliquoted into two fractions. Total RNA extraction from one fraction of transfectants, cDNA synthesis and relative qPCR were performed as described in 'Determination of kinetic class of VZV transcripts' section using the FavorPrep Blood/Cultured Cell Total RNA Mini Kit (FAVORGEN BIOTECH) instead of NucleoSpin RNA. The primer sets for VZV ORF62 (ORF62F2016 and ORF62R2083) and VLT (VLTexon2F and VLTexon2R-2) are presented in Supplementary Table 5. Immunoblotting was performed using a rabbit polyclonal antibody against VZV IE61 or IE63, mouse monoclonal antibodies against alpha-tubulin (B-5-1-2; Sigma-Aldrich) and VZV IE62 (clone 2-B; generous gift from Dr. Yasuyuki Gomi (Research Foundation for Microbial Diseases, Osaka University))[66] or DYKDDDDK (FLAG) tag (WAKO) for pVLT detection as described previously[62]. Uncropped images of immunoblotting are shown in Supplementary Figure 13.

**Data availability.** All sequencing runs were performed using an Illumina NextSeq 550 and all demultiplexed fastq data set are available via the European Nucleotide Archive under study accession number PRJEB23238. RACE-derived VLT$_{ly}$ sequences are available through Genbank under accession numbers MG191301–MG191319.

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

## Acknowledgements

D.P.D. was supported by a New Investigator Award from the Medical Research Foundation (UK MRC) and a small grant provided by the Daiwa Foundation. J.B. was partially funded by the UCL/UCLH Biomedical Research Centre (BRC). T.S. received funding from the Takeda Science Foundation, the Japan Foundation for Pediatric Research, Japan Society for the Promotion of Science (JSPS KAKENHI JP17K008858) and the Ministry of Education, Culture, Sports, Science and Technology (MEXT KAKENHI JP17H05816) and was, in part, supported by a Grant-in-Aid for Scientific Research on Innovative Areas from MEXT of Japan (JP16H06429 and JP16K21723). W.J.D.O., R.J.C. and G.M.G.M.V. were partly supported by National Institutes of Health grant AG032958. R.J.C. was additionally supported by Public Health Service grant NS082228. We acknowledge support from the Medical Research Council and BRC for the UCL/UCLH Pathogen Sequencing Pipeline as well as the UCL Legion High Performance Computing Facility, and associated support services, in the completion of this work. The authors would like to acknowledge Sarah Getu and Suzanne van Veen for technical assistance (Department of Viroscience, Erasmus MC, Rotterdam, The Netherlands) and the whole team of the Netherlands Brain Bank (www.brainbank.nl) for their work and contributions.

## Author contributions

D.P.D., W.J.D.O., T.S., G.M.G.M.V. and J.B. designed the study. D.P.D., W.J.D.O., T.S. and S.E.B. performed experiments. All authors analysed and critiqued the data. D.P.D., W.J.D.O., T.S., R.J.C., G.M.G.M.V. and J.B. wrote the manuscript.

## Additional information

**Competing interests:** The authors declare no competing interests.

