## [Peer Review File(PDF 202 kb) · Nature Communications]

Editorial Note: this manuscript has been previously reviewed at another journal that is not operating a transparent peer review scheme. This document only contains reviewer comments and rebuttal letters for versions considered at *Nature Communications*.

Reviewers' comments:

Reviewer #2 (Remarks to the Author):

In this article to nature communications, Depledge et al report the identification of a transcript that maps antisense to ORF61 in the trigeminal ganglia of human cadavers harboring latent VZV. The significance of the work stems from the difficulty of the systems that have been available to do these types of studies, and the rather messy preceding work that has come before, that has quite befuddled the field. Indeed this group took extraordinary steps to complete the work, and are to be congratulated. While it now looks like VZV is more like its close cousins, it has an impact in that it indicates that VZV is not the "Orphan" it was thought to be. Indeed, I feel a missed impact from the authors point of view is that the conservation across different viruses has more importance, and the fact that these "anti ICP0" transcripts are conserved suggests they have a real role, despite it remaining a frustrating field as to what that real role is- it is still a controversial area in HSV-1, where models and systems (and people working on it) are abundant. Nevertheless, it is this reviewer's opinion that this work is highly important for the VZV field and will have an impact in the herpesvirus field. Most of the work seems well done although there are a few caveats, and a couple of previous issues that are not adequately addressed. Stats and rigor seem to be appropriate.

1. BUT the authors need to back off the "repression of ORF61" claim, they are just not well substantiated in this paper data. The early claims that HSV LATS affected ICP0 expression using similar approaches were all later refuted by mutant and in more viral context studies, and the activity of HSV LAT on ICP0, even through miRNAs, is quite debatable And still under high controversy (and perhaps, still remains there). The influence of VLT on expression of ORF61 is not shown in vivo in VZV infections or in latency, and requires overexpression studies with mutants which are, AS pointed out by 1 and 2 of the prior reviewers, a considerable caveat to the claim. The ratio of one RNA to another RNA in natural circumstances is just one of the variables of whether a RNA actually represses another RNA and protein expression, and this is certainly not going to be represented by doing these types of simple transfection overexpression experiments. In particular, this reviewer takes particular note that the paper TITLE should definitely not indicate this, it is too misleading for the field. I suggest the title be "A novel spliced latency associated VZV transcript mapping antisense to the viral transactivator gene 61". Same goes for line 38-
2. Line 59 ORF63 has been extensively reported as a "latency associated transcript", including by some authors of this work, and is indeed verified here. So line 59 is not correct as it is stated- there are previous reports of VZV LATS
3. Line 155 what conc of pfa is used?
4. The rebuttal indicates that changes were made, but the manuscript does not contain those changes, at least not to the point of improving it. Specifically,
 - a. The figure 3b (and in supp Fig 12c) IF images of VLT does not convey the message very convincingly. The quality of the images is not convincing, despite two reviewers request for

better quality images, and the claim that they were fixed in the rebuttal. Overlap of vLT and VZV proteins is just not clear, and certainly does not indicate cellular distributions and co-distributions. While 3a is better 3b is not. I would have thought that the authors could provide much better data of the expression of the vlat protein than that shown, particularly given the possible importance of the work.

b. There seems to be a stated magnification error, as 3a is far more magnified than 3b, despite an additional "2X digital magnification" . Nuclei are large than the 5uM bar in 3a, while whole cells are much much smaller than the 10uM bar in 3b. Something is amiss.

c. The request for ORFs 54-62 was agreed on by the authors in the rebuttal but is missing from the figure (now 2c)

d. Fig 2 legend says MeWo cells, line 440 text say ARP19 (115, 117). Clarify

Reviewer #3 (Remarks to the Author):

The authors have adequately addressed concerns raised after the initial submission

Comments from the reviewers

In this article to nature communications, Depledge et al report the identification of a transcript that maps antisense to ORF61 in the trigeminal ganglia of human cadavers harboring latent VZV. The significance of the work stems from the difficulty of the systems that have been available to do these types of studies, and the rather messy preceding work that has come before, that has quite befuddled the field. Indeed, this group took extraordinary steps to complete the work, and are to be congratulated. While it now looks like VZV is more like its close cousins, it has an impact in that it indicates that VZV is not the "Orphan" it was thought to be. Indeed, I feel a missed impact from the authors point of view is that the conservation across different viruses has more importance, and the fact that these "anti ICP0" transcripts are conserved suggests they have a real role, despite it remaining a frustrating field as to what that real role is- it is still a controversial area in HSV-1, where models and systems (and people working on it) are abundant. Nevertheless, it is this reviewer's opinion that this work is highly important for the VZV field and will have an impact in the herpesvirus field. Most of the work seems well done although there are a few caveats, and a couple of previous issues that are not adequately addressed. Stats and rigor seem to be appropriate.

1. BUT the authors need to back off the "repression of ORF61" claim, they are just not well substantiated in this paper data. The early claims that HSV LATS affected ICP0 expression using similar approaches were all later refuted by mutant and in more viral context studies, and the activity of HSV LAT on ICP0, even though miRNAs, is quite debatable. And still under high controversy (and perhaps, still remains there). The influence of VLT on expression of ORF61 is not shown in vivo in VZV infections or in latency, and requires overexpression studies with mutants which are, AS pointed out by 1 and 2 of the prior reviewers, a considerable caveat to the claim. The ratio of one RNA to another RNA in natural circumstances is just one of the variables of whether a RNA actually represses another RNA and protein expression, and this is certainly not going to be represented by doing these types of simple transfection overexpression experiments. In particular, this reviewer takes particular note that the paper TITLE should definitely not indicate this, it is too misleading for the field. I suggest the title be "A novel spliced latency associated VZV transcript mapping antisense to the viral transactivator gene 61". Same goes for line 38-

Authors' response: We thank reviewer #2 for their comments and agree that further work is needed to determine whether our observation that VLT inhibits ORF61 transcription in transfected cells also holds true for in vitro models for VZV latency and reactivation. We have toned down the "repression of ORF61" claim in the Abstract (see lines 38-39) and Results/Discussion section (see lines 199-200) and included the reviewer's suggestions on follow-up experiments to assay the role of VLT on ORF61 infection and latency using VLT mutant strains (see lines 200-201) and also adopted the suggested change in title of the revised manuscript. We have also modified the titles of the new Figure 6 (see lines 526-527), and Supplementary Figure 11 (see lines 647-648) appropriately.

2. Line 59 ORF63 has been extensively reported as a "latency associated transcript", including by some authors of this work, and is indeed verified here. So line 59 is not correct as it is stated- there are previous reports of VZV LATS

Authors' response: We have modified the sentence (see lines 60-61) to emphasize that no latency transcripts located antisense to ICP0 have previously been identified. We agree with the reviewer that it is notable that VLT and ORF63 RNA are simultaneously expressed in most latently VZV-infected human TG analyzed, and speculate that both viral transcripts are potentially important for VZV latency or reactivation (see lines 185-187).

3. Line 155 what conc of pfa is used?

Authors' response: The concentration of PFA is 200 µg/mL. This is now included in the Methods section in the revised manuscript in line 883.

4. The rebuttal indicates that changes were made, but the manuscript does not contain those changes, at least not to the point of improving it.

Authors' response: Although we hope that we included the majority of changes requested, we apologize as we did unintentionally miss some out – this has been rectified in the revised manuscript.

Specifically,

a. The figure 3b (and in supplementary Fig 12c) IF images of VLT does not convey the message very convincingly. The quality of the images is not convincing, despite two reviewers request for better quality images, and the claim that they were fixed in the rebuttal. Overlap of vLT and VZV proteins is just not clear, and certainly does not indicate cellular distributions and co-distributions. While 3a is better 3b is not. I would have thought that the authors could provide much better data of the expression of the vlat protein than that shown, particularly given the possible importance of the work.

Authors' response: We had misunderstood the earlier comments from this reviewer and apologize for this. We now include more appropriate pictures of additional experiments to demonstrate pVLT expression in both VLT-transfected and lytically VZV-infected cells (novel Fig. 5a and b). We agree with the reviewer that our data do not show co-localization of pVLT with the VZV proteins gE or ORF62 and consequently deleted the respective conclusions in the revised manuscript text (see lines 152-154 and 171), modified the title of the new Figure 5 (see line 506), and Supplementary Figure 10 (see lines 642-644) accordingly. Furthermore, we have included the control staining performed on un-transfected and uninfected cells (new Fig. 5b) that was previously shown in the original Supporting Figure 12.

b. There seems to be a stated magnification error, as 3a is far more magnified than 3b, despite an additional “2X digital magnification”. Nuclei are large than the 5µM bar in 3a, while whole cells are much smaller than the 10µM bar in 3b. Something is amiss.

Authors' response: We do not agree with the reviewer. There is no magnification error. The correct magnifications (and size of the scale bars) are indicated in the figure legend for revised Figures 3a and 3b (now Figures 5a and 5b). ARPE-19 cells, and their nuclei, vary in size and cell size is further affected by transient transfection or VZV infection, which may result in apparent differences in magnification used.

c. The request for ORFs 54-62 was agreed on by the authors in the rebuttal but is missing from the figure (now 2c)

Authors response: We apologize for this oversight. The new Figure 4c has now been updated as requested and we have additionally indicated the ORF61 gene in the new Figure 4b.

d. Fig 2 legend says MeWo cells, line 440 text says ARP19 (115, 117). Clarify

We thank the reviewer for pointing this out. We observed additional upstream exons during lytic transcription of VLT in both ARPE-19 (new Fig. 2) and MeWo cells (new Fig. 4) and have clarified this in the text of the revised manuscript: see lines 117 and 121.

REVIEWERS' COMMENTS:

Reviewer #2 (Remarks to the Author):

the manuscript looks pretty good to me. now is a nice contribution and will be important to the field